# Development of a novel risk score for diagnosing urinary tract infections: Integrating Sysmex UF-5000i urine fluorescence flow cytometry with urinalysis

Vo Anh Vinh Trang[1,2¶], Thien Tan Tri Tai Truyen[3,4¶]*, Minh Thuan Nguyen[4], Huu Phong Mai[4], Tri Cuong Phan[4], Son Hoang Phan[4], Han My Le Nguyen[4], Huong-Dung Thi Nguyen[4], Nguyen Hai Dang Le[1,2], Man Nhi Tu[1], Vo Thanh Vi Huynh[1], Hoang Tram Anh Nguyen[1], Dac Bao Han Ho[1], Ngoc Thuy Uyen Tran[1], Nguyen Ha Uyen Tran[1], Bich-Nhat Thi Le[1], Duc Tuan Doan[1], Huu Doan Pham[1,2], Truong Bao Phan[1], Phu Phat Pham[1], Tuan Vinh Nguyen[1,5], Phuc Cam Hoang Nguyen[1,2]

**1** Binh Dan Hospital, Ho Chi Minh, Vietnam, **2** Faculty of Medicine, Pham Ngoc Thach University of Medicine, Ho Chi Minh, Vietnam, **3** School of Medicine, Tan Tao University, Long An, Vietnam, **4** Faculty of Medicine, Nam Can Tho University, Can Tho, Vietnam, **5** University of Health Sciences, Vietnam National University Ho Chi Minh City, Ho Chi Minh, Vietnam

¶ These authors contributed equally to this work.
* taitruyenmd@gmail.com

## Abstract

### Background

Urinary tract infections (UTIs) are common globally, and are developing increased antibiotic resistance. Despite being the diagnostic "gold standard," urine culture is limited by slow results and a high rate of false negative findings, leading to treatment delays, higher costs, and overuse of empirical antibiotics. Our study aims to develop a rapid and reliable model to predict clinical outcomes.

### Methods

From January 1st to October 31st, 2023, we enrolled patients with symptoms suggesting UTI from the Outpatient Department of our hospital. Inclusion criteria were patients aged ≥18, initially diagnosed with UTI, available urinalysis, flow cytometry, and urinary culture. Exclusion criteria included failed sample collection and cultures, and pregnant women. A case-control study was conducted, with UTI cases defined as ≥ 10^5 CFU/µL and controls as < 10^5 CFU/µL, matched for age and sex in a 1:1 ratio. For validation, retrospective cases from July to December 2022 were selected with matching controls. Using urine culture as the gold standard, the predictive model was developed with backward stepwise logistic regression. Model discrimination was assessed using area under the curve (AUC).

**Data availability statement:** All relevant data are within the manuscript and its Supporting Information files.

**Funding:** The author(s) received no specific funding for this work.

**Competing interests:** The authors have declared that no competing interests exist.

## Results

In our discovery cohort, we included 1,335 UTI cases and 1,282 non-UTI controls, with mean ages of 52.9 ± 17.1 years and 51.9 ± 16.4 years, and females of 76.9% and 77.7%. Using 100 cells/uL as a threshold, bacterial counts demonstrated a sensitivity of 91.0% and specificity of 45.7%. Our novel UTIRisk score, developed from urinalysis and flow cytometry parameters, showed strong discrimination for UTI, with a AUC of 0.82 (95% CI: 0.81–0.84). In the validation cohort, the AUC was 0.77 (95% CI: 0.74–0.80). The UTIRisk score exhibited excellent specificity (96.5%) and high positive predictive value (92.6%). The score performed strongly across subgroups, particularly in males and patients aged ≥65.

## Conclusions

Our UTIRisk score can improve diagnosis, reduce unnecessary urine cultures, optimize antibiotic use, and help control antibiotic resistance in LMICs. Multicenter, and intervention-based studies are warranted before clinical implementation.

## Introduction

Urinary tract infections (UTIs) are among the most prevalent infections worldwide, affecting both community and healthcare settings, with over 400 million cases annually. They are responsible for more than 236,000 deaths each year, with mortality rates on the rise [1]. Additionally, the growing challenge of antibiotic resistance (ABR) in recent years has exacerbated the difficulty in managing UTI, particularly in developing countries. Gram-negative bacteria, including *Escherichia coli*, have become a major concern due to their production of extended-spectrum β-lactamases (ESBLs) which may increase mortality 3.5-fold [2,3]. Specifically, *E. coli* resistance to cephalosporins ranges from 54% to 66%, while resistance to beta-lactams falls between 26% and 35% [4]. Moreover, significant yearly increases in resistance to carbapenem antibiotics have been observed, further complicating treatment strategies [4]. This rising trend of ABR poses significant challenges, placing additional burden on healthcare systems in affected developing countries, where it has been linked to a 58% rise in crude mortality rates and a 96% increase in ICU admissions [5]. ABR is not restricted to low- and middle-income countries (LMICs); it is also an escalating issue in high-income nations, with projected death rates expected to reach 90.5 per 100,000 persons by 2050 [6]. In Canadian hospitals, the prevalence of ESBL-producing *E. coli* and *K. pneumoniae* rose from 3.4% to 7.1% and 1.5% to 4.0%, respectively, between 2007 and 2011 [7]. Moreover, in these countries, ABR has been associated with an 84% increase in mortality, an additional 7.4 days of hospital stay, a 49% higher rate of readmissions, and extra costs ranging from US$2,300 to US$29,000 [8].

According to the guidelines of the European Association of Urology (EAU), urine culture is regarded as the "gold standard" for diagnosing UTI [9]. While this method enables accurate identification of the causative agent and its antibiotic sensitivity,

it comes with notable challenges. Firstly, urine cultures require highly specialized skills and training, making them labor-intensive processes. Secondly, the turnaround time for results is slow, typically 2–5 days, which delays timely treatment decisions. Consequently, clinicians often resort to empirical treatment with broad-spectrum antibiotics, leading to overprescription and further enabling ABR [10]. These disadvantages and the time and resources required also drive up costs for clinical microbiology laboratories. Unnecessary urine cultures with negative results can lead to a waste of over $10 per sample, accumulating to more than $5,000 annually in a tertiary hospital [11,12].

Recent studies have demonstrated the effectiveness of flow cytometry as a diagnostic tool for UTIs with a sensitivity of over 90% and a specificity ranging from 50% to 84% [13–19]. However, studies integrating flow cytometry into a comprehensive risk score to improve clinical practice are lacking, especially in developing countries where infectious diseases remain a significant challenge. Therefore, we conducted this study to address the absence of a UTI risk score for initial diagnosis, which could help minimize unnecessary urine cultures, reduce patient costs, and, most importantly, provide clinicians with a valuable tool for timely UTI diagnosis. This, in turn, may help prevent risks associated with delayed diagnosis, such as treatment failure and severe complications, while also reducing the overuse of empirical antibiotics and mitigating the threat of ABR. Our objectives were: (1) to assess the efficacy of flow cytometry in UTI diagnosis and (2) to develop a risk score that integrates flow cytometry with urinalysis—an efficient and widely accessible test—to enhance early UTI detection.

## Materials and methods

### Study design and population selection

From January 1st, 2023 to October 31st 2023, we prospectively enrolled patients admitted to our Outpatient Department with symptoms suggesting UTI. The inclusion criteria were defined as patients aged >18 with initial diagnosis of UTI by our physicians based on clinical presentation in the outpatient clinics, with indication of urinary analysis and flow cytometry, and having urinary culture results. The exclusion criteria were defined as patients with failed sample collection or collection without enough data (flow cytometry) to analyze, failed culture, and self-reported pregnancy in women. We conducted a case-control study from which participants were selected in this population. UTI cases were defined if patients had a bacterial count greater than $10^5$ CFU/uL [9]. Controls were defined as patients without a UTI diagnosis, characterized by a bacterial count < $10^5$ cells per μL. Controls were randomly selected from the same study population and matched to UTI cases in a 1:1 ratio based on age and sex to ensure demographic comparability. The matching process was performed using the case-control matching function in SPSS (IBM).

To validate our results, ensuring that the validation cohort comprised at least 30% of the discovery cohort, we retrospectively selected the UTI cases who were diagnosed in our Outpatient Department from July 1st 2022 to December 31st 2022. UTI diagnosis was defined using urinary culture result from medical records during that period. Two physicians (V.A.V.T. and P.P.P) reviewed all the chart in that time to select the UTI cases. The control group comprised patients admitted to the hospital with symptoms suggestive of a UTI but whose urine culture results were negative during the same time period. Controls were matched to cases by age and sex, in a 1:1 ratio to ensure comparability between the groups.

### Sample collection

Urine samples were collected by obtaining 20–30 ml of midstream urine by discarding a small amount of the initial urine and then collecting the following flow in a sterile container. Samples were immediately sent to the laboratory receiving area, ensuring delivery within 2 hours of collection. The ideal time for sample collection was early morning, as bacteria in the bladder have had time to multiply overnight. Samples were not collected if the patient was menstruating or consuming alcohol or other stimulants. This was to avoid potential bias due to the significant variation in erythrocyte and epithelial cell counts during menstruation, as well as the possible changes in pH and specific gravity caused by alcohol and stimulants [20,21]. The collected samples were subsequently analyzed through standard urinalysis, Gram staining, urine culture, and Sysmex UF-5000 analysis.

## Variable definition and collection

Urinalysis was performed using dipstick testing, measuring the following variables: urine concentrations of glucose, ketones, bilirubin, protein, nitrite, leukocytes, and red blood cells (RBC), along with pH and specific gravity.

Flow cytometry analysis of urine was performed using the Sysmex UF-5000i, a third generation fully automated flow cytometry analyzer developed by Sysmex Corporation (Kobe, Japan). Following variables were obtained and included in the analysis: RBC, white blood cell (WBC), WBC clumps, squamous epithelial cells, non-squamous epithelial cells, transitional epithelial cells, renal tubular epithelial cells, hyaline casts, non-hyaline casts, bacterial cells number, yeast-like-cell, and UTI flag by the machine. A detailed definition of each variable was included in supplemental materials (S1 Table). Gram staining and urine culture results were also included.

## Statistical analysis

Bacterial urine culture results were considered the gold standard for UTI diagnosis. Categorical variables were reported as frequencies, while continuous variables were reported as means +/- standard deviations or medians with interquartile ranges, depending on the normality of the distribution assessed using the Kolmogorov–Smirnov test. Univariable analyses were performed to identify significant differences between UTI cases and non-UTI controls. The Chi-square test was used for categorical variables, while the student's t-test was applied to normally distributed continuous variables, and the Mann–Whitney U test for non-normally distributed variables. Only binary variables were included in the predictive model to ensure clinical utility. For continuous variables, we used receiver operating characteristic (ROC) analysis to identify optimal thresholds based on the highest Youden's index (calculated as $J = sensitivity + specificity - 1$) [22,23], converting them into binary variables. All significant variables identified in univariable analysis were included in model development. Due to potential Type I errors arising from multiple comparisons, we adjusted the significance threshold using the Holm-Bonferroni correction ($\alpha adjusted = \frac{\alpha}{n-i+1}$), where $n$ represents the total number of tests (9 for urinalysis variable comparisons and 12 for flow cytometry variable comparisons), and $i$ denotes the rank of the p-value [24]. Variables with missing data were excluded from the model development to ensure the robustness and accuracy of the results. The prediction model was constructed using backward stepwise logistic regression. The regression selection process began with all variables included in the initial model. Each predictor was evaluated for significance, and those with a p-value above 0.05 (the threshold for statistical significance) were excluded. A new model was then tested without the removed variable, and this process was repeated iteratively until all remaining variables had a p-value below 0.05. By incorporating all variables in the same model, the predictive power of each was adjusted, allowing the least significant variable to be eliminated. This approach helped minimize redundancy and reduce variance inflation within the model. Odds ratios from the final model were used as weights to formulate the UTIRisk score for UTI prediction, rounded to one decimal place. Model discrimination was assessed using ROC curves, with the area under the curve (AUC) (C statistic) serving as a performance measure. Quartile-based risk stratification was applied to compare risks, with the first quartile as the reference. Odds ratios were calculated to assess differences across quartiles. Acknowledging that female sex and older age are important risk factors for UTI, we compared the performance of the UTIRisk score across subgroups based on sex (male vs. female) and age (<65 vs. ≥65) to assess the consistency of the UTIRisk score in subgroups with different characteristics. The AUCs for these groups were illustrated with the pROC package in R. The UTIRisk score was validated in a separate cohort following the same methodology. All statistical analyses were conducted using SPSS, and a two-tailed p-value < 0.05 was considered statistically significant.

## Ethical consideration

The study was reviewed approved by the ethical committee of Binh Dan Hospital (Decision number 936/QD-BVBD with specific code CS/BVBD/22/12). The need to obtain consent for the study was waived by the committee. All the urine samples were analyzed for possible UTI as ordered by the clinicians. Patient's data were deidentified before analysis.

## Inclusivity in global research

Additional information regarding the ethical, cultural, and scientific considerations specific to inclusivity in global research is included in the Supporting Information (S2 Checklist)

## Results

### Study population and general characteristics

In our discovery cohort, we included 1,335 UTI cases and 1,282 non-UTI controls, with a mean age of 52.9 ± 17.1 years and 51.9 ± 16.4 years. Cases included 1,026 (76.9%) females, and controls included 996 (77.7%) females in the analysis (Fig 1 and Table 1). The validation cohort consisted of 472 UTI cases and 436 non-UTI controls, with mean ages of 51.1 ± 17.0 and 49.6 ± 15.9 years, and 394 (83.5%) and 374 (85.8%) females (Table 2).

### Urinalysis and flow cytometry results of UTI cases and non-UTI controls

UTI cases showed a lower specific gravity compared to non-UTI controls (1.0145 ± 0.0068 vs. 1.0152 ± 0.0074, p = 0.01). They also had higher percentage of leukocyturia, hematuria, proteinuria, and positive results for nitrites and ketones compared to non-UTI controls (Table 1). Concerning flow cytometry characteristics, UTI cases presented with elevated counts of RBCs, WBCs, WBCs clump, non-hyaline casts, transitional epithelial cells, yeast-like-cell, and particularly bacterial cells in the urine, relative to non-UTI controls (Table 3). Conversely, UTI cases exhibited lower counts of non-squamous epithelial cells and renal tubular epithelial cells than non-UTI controls.

### Diagnostic value of urinalysis and flow cytometry results

For urinalysis results, a positive nitrite test showed high specificity (95.0%) and moderate sensitivity (43.8%) for diagnosing UTI, with an unadjusted odds ratio (OR) of 14.8 (95% CI: 11.3–19.5). In contrast, a positive leukocyte test demonstrated a specificity of 39.9% and high sensitivity of 84.9%, with an unadjusted OR of 3.7 (95% CI: 3.1–4.5). Flow cytometry UTI flag results showed good sensitivity at 92.1% but low specificity of 31.0%, with a corresponding unadjusted OR of 5.2 (95% CI: 4.1–6.6). The C-statistics for these diagnostic markers were 0.69 (95% CI: 0.67–0.71), 0.62 (95% CI: 0.60–0.65), and 0.62 (95% CI: 0.59–0.64), respectively. Additionally, flow cytometry measurements of WBCs count, and bacterial presence

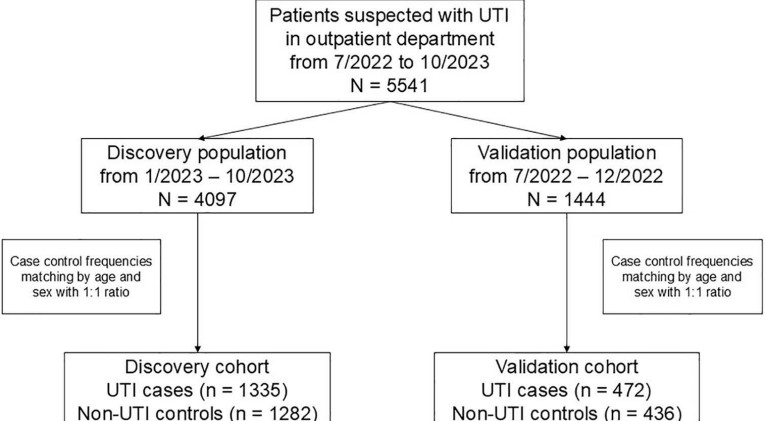

**Fig 1. The flow chart of sample selection.** The discovery cohort included 4,097 outpatients with suspected UTIs prospectively enrolled from January to October 2023. 1,335 (32.6%) outpatients were defined as UTI cases, with a urine culture ≥10⁵ CFU/μL. Cases were matched 1:1 by age and sex with 1,282 non-UTI controls, resulting in a final discovery cohort. The validation cohort was similarly constructed from retrospectively collected data from 7/2022 to 12/2022.

**Table 1. Demographic characteristics and urinalysis results of urinary tract infection cases and controls in discovery cohort.**

| | UTI cases (n = 1335) | Non-UTI controls (n = 1282) | p-value |
|---|---|---|---|
| Age, mean ± SD[a], year | 52.9 ± 17.1 | 51.9 ± 16.4 | 0.11 |
| Female, n (%) | 1,026 (76.9) | 996 (77.7) | 0.61 |
| Urinalysis results | | | |
| Specific gravity, mean ± SD[a,b] | 1.0145 ± 0.0068 | 1.0152 ± 0.0074 | 0.01 |
| Leukocyturia, n % | 1,134 (84.9) | 771 (60.1) | <0.001 |
| Hematuria, n % | 565 (64.8) | 419 (43.9) | <0.001 |
| Missing[d] | 463 | 328 | |
| Nitrite positive, n % | 585 (43.8) | 64 (5.0) | <0.001 |
| pH, mean ± SD | 6.4 ± 0.8 | 6.3 ± 0.8 | 0.10 |
| Glucosuria positive, n % | 56 (4.2) | 52 (4.1) | 0.86 |
| Proteinuria, n % | 488 (36.6) | 313 (24.4) | <0.001 |
| Bilirubin, n %[c] | 2 (0.1) | 0 (0.0) | 0.50 |
| Ketone positive, n % | 18 (1.3) | 5 (0.4) | 0.01 |

[a]SD: Standard deviation.

[b]Number was expressed with 4 decimal numbers due to relatively small value

[c]Fisher exact test

[d]The frequency was calculated with denominator not including missing values

**Table 2. Demographic characteristics and urinalysis results of urinary tract infection cases and controls in validation cohort.**

| | UTI cases (n = 472) | Non-UTI controls (n = 436) | p-value |
|---|---|---|---|
| Age, mean ± SD[a], year | 51.1 ± 17.0 | 49.6 ± 15.9 | 0.15 |
| Female, n (%) | 394 (83.5) | 374 (85.8) | 0.34 |
| Urinalysis results | | | |
| Specific gravity, mean ± SD[a,b] | 1.0147 ± 0.0074 | 1.0145 ± 0.0076 | 0.71 |
| Leukocyturia, n % | 386 (81.8) | 255 (58.5) | <0.001 |
| Hematuria, n % | 223 (55.8) | 137 (34.3) | <0.001 |
| Missing[d] | 72 | 36 | |
| Nitrite, n % | 181 (38.3) | 17 (3.9) | <0.001 |
| pH, mean ± SD | 6.3 ± 0.8 | 6.2 ± 0.8 | 0.13 |
| Glucosuria positive, n % | 16 (3.4) | 16 (3.7) | 0.82 |
| Proteinuria, n % | 131 (27.8) | 80 (18.3) | 0.001 |
| Bilirubin, n (%)[c] | 0 (0.0) | 0 (0.0) | 1.00 |
| Ketone positive, n % | 18 (1.3) | 5 (0.4) | 0.009 |

[a]SD: Standard deviation.

[b]Number was expressed with 4 decimal numbers due to relatively small value

[c]Fisher exact test

[d]The frequency was calculated with denominator not including missing values

**Table 3. Flow cytometry features of urinary tract infection cases and controls in discovery cohort.**

| | UTI cases[a] (n = 1335) | Non-UTI controls (n = 1282) | p-value |
|---|---|---|---|
| Number of red blood cell in urine, median (IQR), cell/uL[b,c] | 17.8 (4.7–86.9) | 12.7 (3.5–70.9) | 0.001 |
| Number of white blood cell in urine, median (IQR), cell/uL | 413.7 (67.6–1,689.7) | 37.2 (7.4–197.8) | <0.001 |
| Number of white blood cell clump in urine, median (IQR), cell/uL | 7.7 (0.8–43.6) | 0.2 (0.0–2.7) | <0.001 |
| Number of hyaline cast in urine, median (IQR), cast/uL | 0.13 (0.0–0.3) | 0.13 (0.0–0.3) | 0.59 |
| Number of non-hyaline cast in urine, median (IQR), cast/uL | 0.13 (0.0–0.4) | 0.0 (0.0–0.1) | <0.001 |
| Number of squamous epithelial cell, median (IQR), cell/uL | 5.3 (1.6–16.7) | 4.8 (1.1–20.8) | 0.29 |
| Number of non-squamous epithelial cell, median (IQR), cell/uL | 1.9 (0.7–5.3) | 2.2 (0.7–6.0) | 0.04 |
| Number of transitional epithelial cell, median (IQR), cell/uL | 0.2 (0.1–1.0) | 0.1 (0.0–0.5) | <0.001 |
| Number of renal tubular epithelial cell, median (IQR), cell/uL | 1.3 (0.2–4.1) | 1.9 (0.5–5.3) | <0.001 |
| Number of bacterial cell in urine, median (IQR), cell/uL | 37,26.6 (589.6–27,273.7) | 126.65 (26.1–767.9) | <0.001 |
| Number of yeast-like-cell, median (IQR), cell/uL | 1.3 (0.5–4.1) | 1.0 (0.2–2.9) | <0.001 |
| UTI flag, n (%) | 1,229 (92.1) | 884 (69.0) | <0.001 |

UTI: Urinary Tract Infection; IQR: Interquartile range; uL = micro liter.

[a]UTI: Urinary Tract Infection

[b]IQR: Interquartile range

[c]uL = micro liter.

demonstrated good UTI discrimination, with C-statistics of 0.73 (95% CI: 0.71–0.75) and 0.82 (95% CI: 0.81–0.84) (Fig 2). Using 100 cell/uL as threshold, bacterial counts demonstrated a sensitivity of 91.0% and specificity of 45.7%.

## Development of UTIRisk score and its predictive value of UTI

Using multivariable logistic regression, we developed an optimal model called the UTIRisk score to predict UTI. This model includes the following variables: urine ketone positivity, absence of proteinuria, nitrite positivity, RBCs counts in urine ≤ 24 cells/μL, WBCs count in urine ≥ 200 cells/μL, WBCs clumps count ≥ 4.2 cells/μL, non-hyaline casts in urine ≤ 1.3 casts/μL, a positive UTI flag, and bacterial count in urine ≥ 455 cells/μL (all determined by flow cytometry). The coefficients for each component are detailed in Table 4.

In the discovery cohort, UTI cases had a higher average UTIRisk score than non-UTI controls (11.2 vs. 5.6, p < 0.001) (Fig 3). The odds of UTI increased by 1.37 (95% CI: 1.33–1.40) for each 1-point increase in the UTIRisk score (Table 5). Patients with UTIRisk scores in the 2nd quartile and above had significantly higher odds ratios for UTI compared to those in the lowest quartile (Table 5). The UTIRisk score demonstrated strong discrimination for UTI, with a C-statistic of 0.82 (95% CI: 0.81–0.84), comparable to the bacterial count in urine (Fig 2). Similar results were replicated in the validation cohort, with an AUC of 0.77 (95% CI: 0.74–0.80) and an odds ratio of 1.37 (95% CI: 1.33–1.40) for each 1-point increase in the UTIRisk score (Table 5 and Fig 2).

Using a diagnostic threshold score of 14, the UTI Risk Score demonstrated excellent specificity and a high PPV (96.5% and 92.6%, respectively). Applying this risk score has the potential to reduce unnecessary urine cultures by 21.6%. Similar results were observed in the validation cohort, with a specificity of 96.8% and a PPV of 92.5%.

## Subgroup analysis

The sex-stratified analysis indicated that the UTIRisk score exhibited superior discrimination performance in male patients compared to female patients, with an AUC of 0.89 (95% CI: 0.86–0.92) versus 0.80 (95% CI: 0.78–0.82). In the

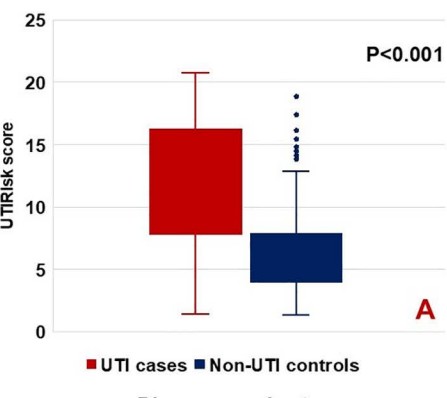
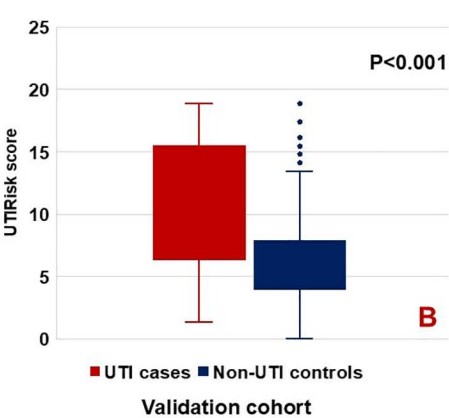

**Fig 2. UTIRisk score of UTI cases and non-UTI controls in discovery and validation cohorts.** In the discovery cohort, UTI cases had a significantly higher median UTIRisk score than non-UTI controls (11.2 vs. 5.6, p<0.001, 2A). Similar results were observed in the validation cohort (9.8 vs. 5.6, p<0.001, 2B).

age-stratified analysis, using 65 years as the threshold, the UTIRisk score demonstrated better performance in patients aged 65 years and older, with an AUC of 0.86 (95% CI: 0.83–0.89) compared to 0.81 (95% CI: 0.79–0.83) for younger patients (Fig 4). Similar sex-disparity was observed in the validation cohort with the AUC for male and female were 0.87 (95% CI: 0.81–0.93) and 0.75 (95% CI: 0.71–0.78).

## Discussion

Our study demonstrated that flow cytometry is promising in diagnosing UTIs among patients with suspected symptoms, exhibiting high sensitivity and moderate specificity. The bacterial count and UTI-flag results obtained through flow cytometry demonstrated strong discriminatory ability in identifying UTI cases. Notably, the newly developed UTIRisk score showed good discriminatory ability for UTI diagnosis in the discovery cohort. This result was successfully replicated in the validation cohort, indicating the robustness and consistency of the risk score. Additionally, this risk score demonstrated a strong overall performance across different subgroups, including sex and age, with particularly notable efficacy in males and patients aged >65.

**Table 4. The components of our novel UTIRisk score.**

| Number | Variable | Component score |
|---|---|---|
| Urinalysis features | | |
| 1 | Urine Ketone positive | 3.2 |
| 2 | Negative proteinuria | 1.4 |
| 3 | Nitrite positive | 6.3 |
| Flow cytometry characteristics | | |
| 4 | The number of red blood cell in urine determined by flow cytometry is less than 24 cell/uL | 1.3 |
| 5 | The number of white blood cell in urine determined by flow cytometry is more than 200 cell/uL | 1.4 |
| 6 | The number of white blood cell clump in urine determined by flow cytometry is more than 4.2 cell/uL | 2.0 |
| 7 | The number of non-hyaline casts in urine determined by flow cytometry is less than 1.3 cast/uL | 1.3 |
| 8 | UTI flag by Sysmex 5000i | 1.6 |
| 9 | The number of bacteria in urine determined by flow cytometry is more than 455 cell/uL | 3.5 |

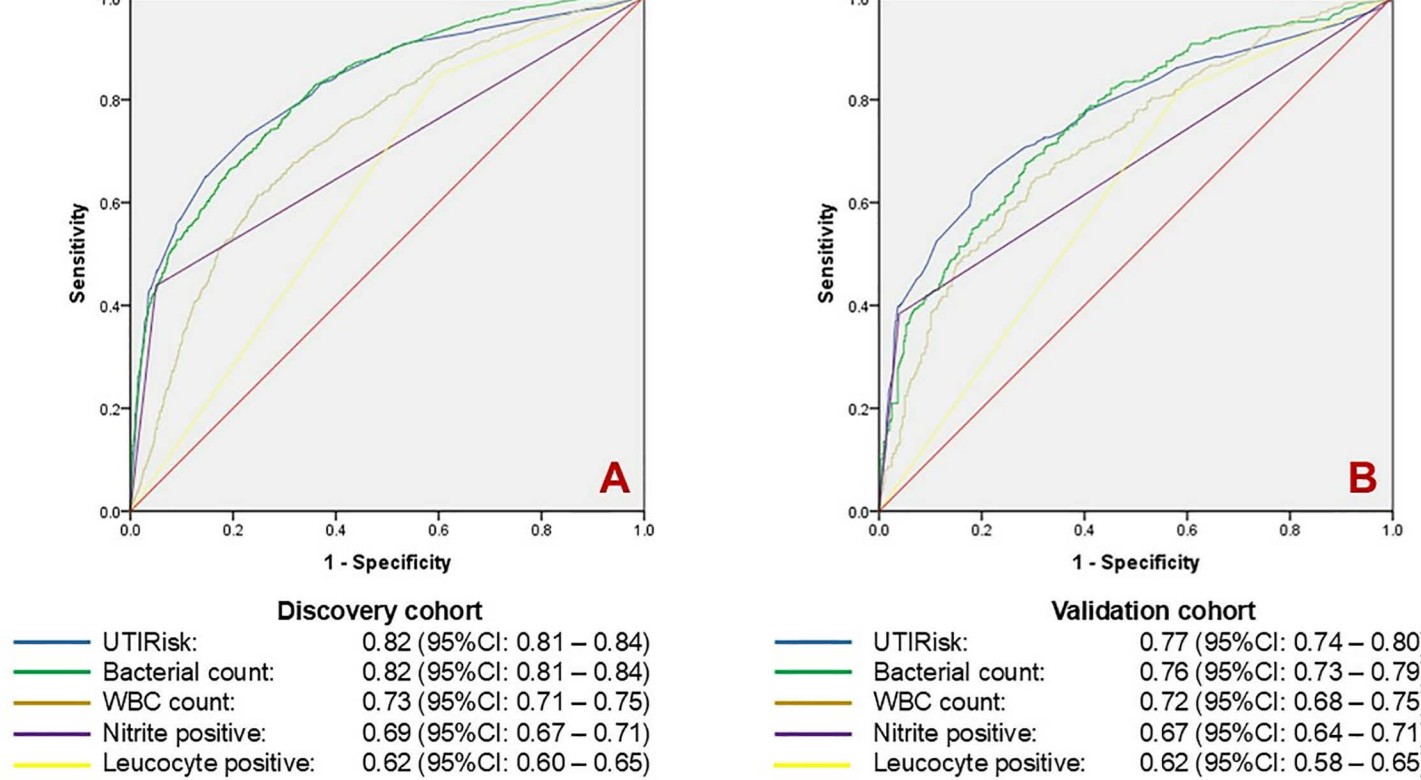

**Fig 3. The UTI discrimination performance of our novel UTIRisk score and other parameters of flow cytometry and urinalysis in discovery and validation cohorts.** The UTIRisk score demonstrated strong discriminatory ability, comparable to bacterial counts from flow cytometry, and outperformed commonly used parameters, including white blood cell WBC count from flow cytometry, nitrite, and leukocyte positivity from urinalysis, in both the discovery (3A) and validation cohorts (3B).

Our study demonstrated that the UTI flag and bacterial count, using a diagnostic threshold of 100 cells/μL, exhibit good sensitivity in diagnosing UTIs. These findings are consistent with a previous study that highlighted the excellent sensitivity of flow cytometry, specifically using the Sysmex UF-5000i as a reliable screening tool for UTI diagnosis [13–17]. On the other hand, the specificity observed in our study (45.7%) was similar to that reported in a Brazilian study (50%) [18] but significantly lower than the specificity reported in other studies, which ranged from 74.6% to 85.4% [13–17]. This discrepancy underscores the disparity in the performance of this diagnostic method between LMICs such as Vietnam and Brazil and more developed nations such as Italy, Austria, Norway, China, and South Korea. This disparity may be attributed to the higher levels of normal flora bacterial colonization [25], particularly Gram-positive bacteria, on the skin and mucosal surfaces, which are associated with poorer socioeconomic conditions commonly found in LMICs [26]. Such colonization can interfere with the accuracy of flow cytometry bacterial counts, potentially leading to an increased rate of false positives. Moreover, our findings revealed a strong discriminatory power of WBCs and bacterial counts measured by flow cytometry for UTI diagnosis, with a similar AUC aligning with previous research. These results reinforce the utility of flow cytometry in diagnosing UTIs and contribute additional evidence of its effectiveness in LMIC settings. Finally, while the high sensitivity of flow cytometry makes it useful for screening, its low specificity can lead to overdiagnosis and unnecessary treatment, especially in LMICs. These results highlight the importance of combining flow cytometry with other diagnostic modalities to enhance its diagnostic accuracy.

**Table 5.  UTIRisk score of urinary tract infection cases and controls in discovery and validation cohorts.**

|  | Values |
| --- | --- |
| **Discovery cohort (n = 2,617)** | |
| *UTIRisk score points, median (interquartile)* | |
| UTI cases | 11.2 (7.8–16.2) |
| Non-UTI controls | 5.6 (4.0–7.8) |
| Odds ratio (95% CI), per 1-unit increase in UTIRisk score | 1.37 (1.33–1.40) |
| *Odds ratio (95% CI), by UTIRisk score quartile* | |
| Quartile 1 (1.3–4.3) | 1 |
| Quartile 2 (4.4–7.8) | 2.1 (1.6–2.8) |
| Quartile 3 (7.9–12.5) | 6.6 (5.0–8.6) |
| Quartile 4 (12.5–20.7) | 53.4 (37.1–77.0) |
| **Validation cohort (n = 908)** | |
| *UTIRisk score points, median (interquartile)* | |
| UTI cases | 9.8 (6.3–15.4) |
| Non-UTI controls | 5.6 (4.0–7.8) |
| Odds ratio (95% CI), per 1-point increase in UTIRisk score | 1.30 (1.25–1.35) |
| *Odds ratio (95% CI), by UTIRisk score quartile* | |
| Quartile 1 (0.0–4.3) | 1 |
| Quartile 2 (4.4–7.8) | 1.4 (0.9–2.1) |
| Quartile 3 (7.9–11.2) | 3.7 (2.5–5.6) |
| Quartile 4 (11.3–18.8) | 27.3 (15.5–47.9) |

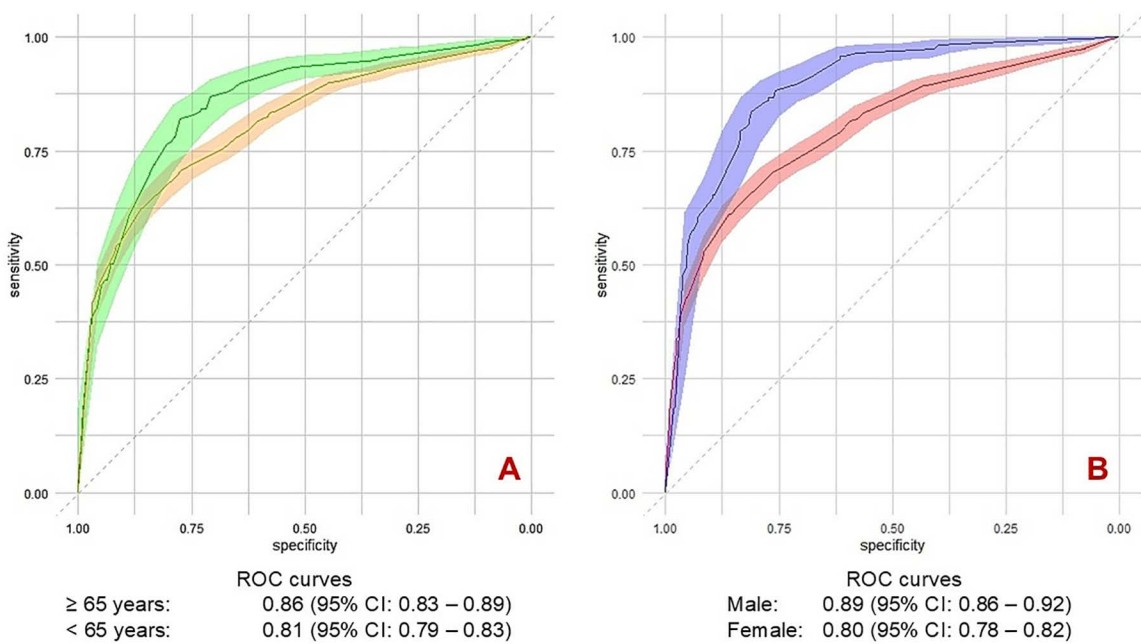

ROC curves
≥ 65 years:    0.86 (95% CI: 0.83 − 0.89)
< 65 years:    0.81 (95% CI: 0.79 − 0.83)

ROC curves
Male:    0.89 (95% CI: 0.86 − 0.92)
Female:  0.80 (95% CI: 0.78 − 0.82)

**Fig 4.  Discrimination performance of UTIRisk score across age and sex subgroups.** The UTIRisk score showed consistently strong diagnostic ability across age and sex subgroups (AUC ≥ 0.80), with excellent performance in patients aged ≥65 (4A) and males (4B).

One of the current challenges in applying flow cytometry to clinical practice is the variation in bacterial count used as a diagnostic threshold across countries and studies. Different thresholds can be employed to improve sensitivity and negative predictive value for screening tools or enhance specificity and positive predictive value for alternative diagnostic methods. To enhance the independence and generalizability of flow cytometry, integrating this laboratory test into a risk score model that includes other laboratory parameters would be an effective solution. Based on that reason, we developed a novel UTIRisk score by combining various parameters from urinalysis and flow cytometry into a 9-component risk score, demonstrating excellent discriminatory power for UTI detection in both the discovery and validation cohorts, with an AUC of 0.82 (95% CI: 0.81–0.84) and an AUC of 0.77 (95% CI: 0.74–0.80), respectively. To our knowledge, this is the first risk score that combines urinalysis and flow cytometry parameters to predict UTI in patient with suggested clinical symptoms. Previous UTI risk scores were mainly developed for hospitalized patients, whose characteristics and pathogens differ from our target population [27–29]. While these models showed strong predictive power (C-statistic: 0.79–0.84), they focused on UTIs acquired during hospitalization, often involving antibiotic-resistant pathogens like multidrug-resistant *Escherichia coli* and *Pseudomonas aeruginosa*. These scores incorporated demographics (age, sex), laboratory results (blood glucose), medical history, and clinical indicators (Glasgow Coma Score, National Institute of Health Stroke Scale, urinary catheter use) and were designed for prediction and prevention. In contrast, our score prioritizes early diagnosis. In contrast, our risk score is primarily intended for early diagnosis. Notably, only one risk score, developed in England for primary care, aligns with our outpatient setting by focusing on early UTI detection using urinalysis variables [30]. With a diagnostic threshold range from 3 to 3.5, this risk score demonstrated comparable specificity and PPV (90–97% and 84–93%, respectively) to the results observed in our study. However, this score used the diagnostic criteria based on the European guidelines at the time, which specified a threshold of $10^3$ CFU/uL. In contrast, our UTIRisk score was developed using the most recent EAU guidelines, with a threshold of $10^5$ CFU/uL, which is more up-to-date and potentially more applicable in clinical scenarios.

Our UTIRisk score incorporated nine components derived from urinalysis and flow cytometry. In addition to traditional parameters for predicting UTIs—such as positive urine nitrite, increased WBC count, the presence of a UTI flag, and the number of detected urinary bacteria—several novel predictors warrant discussion. First, the association between ketonuria and infection has been well established [31]. Ketone bodies are an alternative energy source during increased fat metabolism in stress scenarios, such as infection. Moreover, ketone bodies have been recognized as critical metabolites in regulating macrophage and T-cell function, which play essential roles in host defense against infection [32]. Second, high cut-off values for urinary protein levels, RBCs, and non-hyaline cast numbers may exclude glomerular and tubular diseases, thereby enhancing the diagnostic accuracy for UTIs. This approach underscores the utility of integrating novel and traditional biomarkers to evaluate UTI risk comprehensively.

Our UTIRisk score has several potential clinical applications, including improving the early diagnosis of UTI, reducing unnecessary urine cultures, and preventing the overuse of antibiotics, which could help mitigate antibiotic resistance, particularly in LMICs. This risk score combines variables from flow cytometry and urinalysis, both of which are quick, easy-to-obtain tests that are inexpensive and require minimal technical expertise. These features enhance diagnostic speed and increase accessibility, making the UTIRisk score particularly valuable in LMICs, where infrastructure and insurance coverage can be challenging. Additionally, the UTIRisk score is user-friendly and can be integrated into mobile applications, allowing physicians to use it quickly and efficiently. Perhaps the most significant advantage of this risk score is that it minimizes reliance on a single diagnostic parameter, such as bacterial count, which can be influenced by microbiome characteristics and socioeconomic factors in different countries. This approach helps address the variability in diagnostic thresholds proposed by previous studies [13–19].

## Strengths and limitations

Our study demonstrated several strengths that deserve discussion. Firstly, the application of flow cytometry in LMICs, particularly in Asia, remains limited. Conducting this study at one of the largest urology hospitals in Vietnam, with a large

sample size, allowed us to analyze both urinalysis and flow cytometry data and to contribute valuable insights not only into the diagnostic value of flow cytometry but also into the broader laboratory characteristics of UTI patients in LMICs. Secondly, to our knowledge, this is the first study to integrate flow cytometry findings into developing a risk score to aid in diagnosing UTIs. Our risk score demonstrated consistent efficacy across various age and sex groups, suggesting broad applicability. Our risk score can potentially reduce unnecessary urine cultures, improve diagnostic efficiency, and enhance the utility of flow cytometry in clinical practice. Lastly, our study design included two case-control cohorts matched by age and sex, reduced bias, and allowed for internal validation. This robust methodological framework enhances the credibility of our findings. However, some limitations should be acknowledged. Our risk score was developed solely based on laboratory findings and did not incorporate clinical parameters or medical history, which could further improve the diagnostic accuracy. Additionally, as a single-center study, despite the large sample size and internal validation, the generalizability of the risk score would benefit from external validation across other medical centers to confirm its reliability in varied settings. Finally, the model was developed using backward stepwise logistic regression, which, while effective in variable selection, does not completely eliminate multicollinearity between variables.

## Conclusion

UTIs are among the most common infectious diseases in LMICs. By integrating flow cytometry results with traditional urinalysis parameters, our novel UTIRisk score can enhance diagnosis, reduce unnecessary urine cultures, improve clinician antibiotic prescribing, and, ultimately, help control the growing issue of ABR, particularly in LMICs. However, further studies with a multicenter design are needed to validate the efficacy of this diagnostic model across diverse populations, including both LMICs as well as high-income developed countries. Additionally, intervention-based studies, such as randomized controlled trials, are essential before this risk score can be fully integrated into clinical practice.

## Supporting information

**S1 Checklist. STROBE checklist.**
(DOCX)

**S2 Checklist. Inclusivity in global research.**
(DOCX)

**S1 Dataset. Discovery cohort.**
(CSV)

**S2 Dataset. Validation cohort.**
(CSV)

**S1 Table. Detailed definition of variables obtained from urine flow cytometry.**
(DOCX)

## Acknowledgments

We express our deepest thanks and appreciation to medical staffs (Phuc Nhan Le Bao, Nha Tran Thanh, Kieu My Phan, Quang Nguyen Thien, Mai Truong Nguyen Xuan, Uyen Tran Nguyen Hoang) for data processing and patients involved in this study.

## Author contributions

**Conceptualization:** Vo Anh Vinh Trang, Thien Tan Tri Tai Truyen, Phuc Cam Hoang Nguyen.

**Data curation:** Tri Cuong Phan, Son Hoang Phan, Han My Le Nguyen, Huong-Dung Thi Nguyen, Nguyen Hai Dang Le, Man Nhi Tu, Vo Thanh Vi Huynh, Hoang Tram Anh Nguyen, Dac Bao Han Ho, Ngoc Thuy Uyen Tran, Nguyen Ha Uyen Tran, Bich-Nhat Thi Le, Duc Tuan Doan, Truong Bao Phan.

**Formal analysis:** Thien Tan Tri Tai Truyen, Tri Cuong Phan, Han My Le Nguyen, Huong-Dung Thi Nguyen.

**Investigation:** Vo Anh Vinh Trang, Thien Tan Tri Tai Truyen, Son Hoang Phan, Nguyen Hai Dang Le, Man Nhi Tu, Vo Thanh Vi Huynh, Hoang Tram Anh Nguyen, Dac Bao Han Ho, Ngoc Thuy Uyen Tran, Nguyen Ha Uyen Tran, Bich-Nhat Thi Le, Duc Tuan Doan, Huu Doan Pham, Truong Bao Phan, Phu Phat Pham.

**Methodology:** Vo Anh Vinh Trang, Thien Tan Tri Tai Truyen, Tuan Vinh Nguyen.

**Project administration:** Tuan Vinh Nguyen, Phuc Cam Hoang Nguyen.

**Supervision:** Tuan Vinh Nguyen, Phuc Cam Hoang Nguyen.

**Validation:** Huong-Dung Thi Nguyen, Phu Phat Pham.

**Visualization:** Huong-Dung Thi Nguyen.

**Writing – original draft:** Vo Anh Vinh Trang, Thien Tan Tri Tai Truyen, Minh Thuan Nguyen, Huu Phong Mai.

**Writing – review & editing:** Thien Tan Tri Tai Truyen, Tuan Vinh Nguyen, Phuc Cam Hoang Nguyen.

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
