## [Decision Letter · Decision Letter 0]

6 Mar 2025

PONE-D-25-03936Development of a Novel Risk Score for Diagnosing Urinary Tract Infections: Integrating Sysmex UF-5000i Urine Fluorescence Flow Cytometry with UrinalysisPLOS ONE

Dear Dr. Truyen,

Thank you for submitting your manuscript to PLOS ONE. After careful consideration, we feel that it has merit but does not fully meet PLOS ONE’s publication criteria as it currently stands. Therefore, we invite you to submit a revised version of the manuscript that addresses the points raised during the review process.

We look forward to receiving your revised manuscript.

Kind regards,

Awatif Abid Al-Judaibi, PhD

Academic Editor

PLOS ONE

Journal Requirements:

2. Please include a complete copy of PLOS’ questionnaire on inclusivity in global research in your revised manuscript. Our policy for research in this area aims to improve transparency in the reporting of research performed outside of researchers’ own country or community. The policy applies to researchers who have travelled to a different country to conduct research, research with Indigenous populations or their lands, and research on cultural artefacts. The questionnaire can also be requested at the journal’s discretion for any other submissions, even if these conditions are not met.  

Please find more information on the policy and a link to download a blank copy of the questionnaire here: https://journals.plos.org/plosone/s/best-practices-in-research-reporting. 

Please upload a completed version of your questionnaire as Supporting Information when you resubmit your manuscript.

3. In the online submission form you indicate that your data is not available for proprietary reasons and have provided a contact point for accessing this data. Please note that your current contact point is a co-author on this manuscript. According to our Data Policy, the contact point must not be an author on the manuscript and must be an institutional contact, ideally not an individual. Please revise your data statement to a non-author institutional point of contact, such as a data access or ethics committee, and send this to us via return email. Please also include contact information for the third party organization, and please include the full citation of where the data can be found.

Reviewers' comments:

Reviewer's Responses to Questions

**Comments to the Author**

1. Is the manuscript technically sound, and do the data support the conclusions?

Reviewer #1: Yes

Reviewer #2: Yes

Reviewer #3: Yes

2. Has the statistical analysis been performed appropriately and rigorously? 

Reviewer #1: Yes

Reviewer #2: Yes

Reviewer #3: Yes

3. Have the authors made all data underlying the findings in their manuscript fully available?

Reviewer #1: Yes

Reviewer #2: Yes

Reviewer #3: Yes

4. Is the manuscript presented in an intelligible fashion and written in standard English?

Reviewer #1: Yes

Reviewer #2: No

Reviewer #3: Yes

5. Review Comments to the Author

Reviewer #1: 1. Clarity of Objectives: The introduction should clearly outline the specific objectives of the study. It would benefit from a more detailed discussion of the rationale behind developing the UTIRisk score and how it addresses existing gaps in UTI diagnosis.

2. Methodology:

- The inclusion and exclusion criteria are well defined; however, it would be beneficial to expand on how matching controls were selected to ensure comparability in the study population.

- Consider providing more details on the flow cytometry parameters utilized in developing the UTIRisk score, as readers may need a clearer understanding of the technical aspects involved.

1. Statistical Analysis:

- The methods employed for backward stepwise logistic regression should provide explicit justification, including checking for multicollinearity among predictors.

- It would strengthen the manuscript if the authors could elaborate on how they handled missing data, if any.

1. Results:

- While the results section does well in presenting findings, including confidence intervals for all AUC values would enhance the robustness of the interpretations.

- A more detailed discussion of limitations related to sensitivity (91.0%) and specificity (45.7%) would be valuable. Especially in quantifying the implications of this specificity on clinical practice.

1. Discussion:

- The discussion could be expanded to include comparisons with existing UTI diagnostic criteria and risk scores.

- Consider adding a section discussing potential clinical implications of implementing the UTIRisk score in low- and middle-income countries (LMICs) specifically.

1. Conclusion:

- The conclusions draw valid implications from the study but should also address the need for further research, possibly emphasizing the types of multicenter studies that would be necessary to validate the UTIRisk score across diverse populations.

1. References: Ensure that all cited research is current and relevant, particularly concerning comparisons with previous risk scoring models.

2. Data Availability: The authors should clarify how data will be made available upon reasonable request, including what specific data sets may be available.

Reviewer #2: Congratulations on your work. I have added my suggestions and comments below.

Abstract

Background:

• The background is concise; however, the study aim is not clearly articulated. I wonder if the overall aim of the study can be stated more explicitly.

Methods:

• The inclusion period is clearly defined. It may be useful to clarify whether the outpatient department was hospital-based or community-based, as this distinction could affect generalizability—for sure this should be consistent with the main text.

Results:

• The demographics are well reported. It may enhance clarity to mention whether there were any significant differences in baseline characteristics between the cases and controls.

• The reported AUC of 0.82 is promising. It might be valuable to briefly add the clinical significance of this value, especially in comparison to existing models or methods.

Introduction:

• Line 72-73: This point introduces the challenge of antibiotic resistance. Consider briefly mentioning other high-income settings where antibiotic resistance is also a growing concern, making the issue more relevant to a wider audience. Additionally, explaining the impact on healthcare systems in developing countries could provide more depth.

• Line 73-74: This is a clear and informative point about antibiotic resistance. It would be helpful to clarify the significance of ESBL production for non-specialist readers or consider providing a brief explanation of why this is particularly problematic for UTI management.

• Line 87: This point adds an important economic perspective. It may be useful to provide a brief comparison of the costs of urine culture versus potential alternatives, helping to contextualize the need for more cost-effective diagnostic methods.

• Line 89-90: This statement reiterates the importance of timely diagnosis. Consider integrating the specific risks associated with delayed diagnosis (e.g., treatment failure, patient morbidity) to further justify the need for an improved diagnostic method.

• Line 93-95: This is a good transition to the study’s rationale. I wonder if you could mention briefly whether any studies have been conducted in high-income countries to compare the performance of flow cytometry in similar settings, which would strengthen the argument for your study.

Methods:

• Line 100-101: The inclusion period is clear, but it may help to specify whether this is a prospective or retrospective enrollment process. The term "continuously enrolled" could be refined to clarify if this refers to a continuous intake of patients or if there was a defined selection process. This can avoid any ambiguity.

• Line 102-104: a minor grammatical issue with "who is" which should be corrected to "who are." Also, the inclusion criteria are generally clear, but it would be helpful to specify if the physicians' diagnoses were based solely on clinical presentation or if further diagnostic tests (e.g., urinalysis) were required. This would clarify the diagnostic approach used.

• Line 107-110: There are some minor grammatical issues ("which the participants were selected" could be rephrased to "from which participants were selected"). The definition of UTI cases could be more precise—consider rewording "the number of colonized bacteria is more than" to "a bacterial count greater than 10^5 CFU/uL."

• Line 111-113: The validation cohort is clearly defined, though "For validation our results" could be rephrased to "To validate our results." This minor adjustment improves readability. Also, I wonder if you could state the rationale behind selecting this particular retrospective period—was it to match seasonality, ensure a certain number of cases, or another reason?

• Line 124-125: The rationale for early morning sample collection is appropriate. However, the exclusion of menstruating patients and those using alcohol or stimulants could benefit from further explanation. For instance, why do these factors interfere with sample integrity, and are there other conditions that could be relevant but were not considered?

• Line 129-131: This list of urinalysis variables is comprehensive. It might be worth briefly explaining why these specific variables were selected for testing, especially in relation to their diagnostic relevance for UTI.

• Line 145-149: One minor suggestion, I wonder if you could mention whether any adjustments were made for multiple comparisons, especially if several tests were conducted simultaneously. This would help address potential Type I errors.

• Line 153-154: It would be beneficial to mention any specific criteria used for selecting variables in the backward stepwise process (e.g., p-value threshold). This adds transparency to the methodology and clarifies the decision-making process.

Discussion

• Lines 232-235: The authors effectively highlight the high sensitivity and moderate specificity of flow cytometry for diagnosing UTIs. I believe It would be helpful to mention how this could impact clinical practice in low-resource settings.

• Lines 245-253: The variability in specificity across different countries (LMICs vs. developed countries) is well addressed. Clarifying why these differences exist (e.g., microbiome factors, socioeconomic conditions) could strengthen the argument.

Reviewer #3: Line 33-34 "Inclusion criteria were patients aged ≥18, initially diagnosed with UTI, available urinalysis, flow cytometry, and urinary culture." Clarify if "initially diagnosed with UTI" refers to clinical diagnosis or initial screening, as this could impact the selection bias.

In line 37, could you specify the selection criteria for these retrospective cases to explain how they align with the study's primary objectives?

Lines 160–161, there was subgrouping based on age and sex. It would be beneficial to explain why these subgroups were chosen and discuss any differential impacts observed in more detail.

6. PLOS authors have the option to publish the peer review history of their article (what does this mean? ). If published, this will include your full peer review and any attached files.

**Do you want your identity to be public for this peer review?** For information about this choice, including consent withdrawal, please see our Privacy Policy .

Reviewer #1: No

Reviewer #2: No

Reviewer #3: **Yes: ** Sumayya Al-Mansur

---

## [Author Response · Author response to Decision Letter 0]

7 Mar 2025

RESPONSE TO REVIEWERS

Reviewer #1

Comment #1: Clarity of Objectives: The introduction should clearly outline the specific objectives of the study. It would benefit from a more detailed discussion of the rationale behind developing the UTIRisk score and how it addresses existing gaps in UTI diagnosis.

Response to comment #1: We agreed and revised the manuscript accordingly.

Manuscript changes:

• Page 5. Introduction. Lines 100-112. “Recent studies have demonstrated the effectiveness of flow cytometry as a diagnostic tool for UTIs with a sensitivity of over 90% and a specificity ranging from 50% to 84%.[13-19] However, studies integrating flow cytometry into a comprehensive risk score to improve clinical practice are lacking, especially in developing countries where infectious diseases remain a significant challenge. Therefore, we conducted this study to address the absence of a UTI risk score for initial diagnosis, which could help minimize unnecessary urine cultures, reduce patient costs, and, most importantly, provide clinicians with a valuable tool for timely UTI diagnosis. This, in turn, may help prevent risks associated with delayed diagnosis, such as treatment failure and severe complications, while also reducing the overuse of empirical antibiotics and mitigating the threat of ABR. Our objectives were: (1) to assess the efficacy of flow cytometry in UTI diagnosis and (2) to develop a risk score that integrates flow cytometry with urinalysis—an efficient and widely accessible test—to enhance early UTI detection.”

Comment #2: Methodology: The inclusion and exclusion criteria are well defined; however, it would be beneficial to expand on how matching controls were selected to ensure comparability in the study population.

Response to comment #2: We agreed and specified how did we selected the controls.

Manuscript changes:

• Page 5-6. Study design and population selection. Lines 123-128. “Controls were defined as patients without a UTI diagnosis, characterized by a bacterial count < 10^5 cells per µL. They were randomly selected from the same study population and matched to UTI cases in a 1:1 ratio based on age and sex to ensure demographic comparability. The matching process was performed using the case-control matching function in SPSS (IBM).”

Comment #3: Consider providing more details on the flow cytometry parameters utilized in developing the UTIRisk score, as readers may need a clearer understanding of the technical aspects involved.

Response to comment #3: We agreed and have provided a detailed definition of each variable obtained from flow cytometry analysis or urine in the supporting information (S Table).

Manuscript changes:

• Page 8. Variable definition and collection. Lines 160-161. “A detailed definition of each variable was included in supplemental materials (S Table).”

Comment #4: Statistical Analysis: The methods employed for backward stepwise logistic regression should provide explicit justification, including checking for multicollinearity among predictors.

Response to comment #4: We agreed and elaborated how the backward stepwise logistic regression worked and its impact on the multicollinearity among predictors. We acknowledge that this regression technique still has limitation for resolving multicollinearity. Therefore, we mentioned about it in the limitation.

Manuscript changes:

• Page 9. Statistical analysis. Lines 183-192. “The regression selection process began with all variables included in the initial model. Each predictor was evaluated for significance, and those with a p-value above 0.05 (the threshold for statistical significance) were excluded. A new model was then tested without the removed variable, and this process was repeated iteratively until all remaining variables had a p-value below 0.05. By incorporating all variables in the same model, the predictive power of each was adjusted, allowing the least significant variable to be eliminated. This approach helped minimize redundancy and reduce variance inflation within the model.”

• Page 23. Strengths and limitation. Lines 423-425.” Finally, the model was developed using backward stepwise logistic regression, which, while effective in variable selection, does not completely eliminate multicollinearity between variables.”

Comment #5: It would strengthen the manuscript if the authors could elaborate on how they handled missing data, if any.

Response to comment #5: We had only one variable with missing data which was hematuria determined by urinalysis. We excluded this variable from model development to avoid potential bias.

Manuscript changes:

• Page 9. Statistical analysis. Lines 181-182. “Variables with missing data were excluded from the model development to ensure the robustness and accuracy of the results.”

Comment #6:1. Results: While the results section does well in presenting findings, including confidence intervals for all AUC values would enhance the robustness of the interpretations.

Response to comment #6: We agreed and have added the 95% CI for all AUC values.

Manuscript changes:

• Page 14. Diagnostic value of urinalysis and flow cytometry results. Lines 258-259. “The C-statistics for these diagnostic markers were 0.69 (95% CI: 0.67 – 0.71), 0.62 (95% CI: 0.60 – 0.65), and 0.62 (95% CI: 0.59 – 0.64), respectively.”

Comment #7: A more detailed discussion of limitations related to sensitivity (91.0%) and specificity (45.7%) would be valuable. Especially in quantifying the implications of this specificity on clinical practice.

Response to comment #7: We agreed and have added a brief discussion about limitation of low specificity.

Manuscript changes:

• Page 18. Discussion. Lines 342-345. “Finally, while the high sensitivity of flow cytometry makes it useful for screening, its low specificity can lead to overdiagnosis and unnecessary treatment, especially in LMICs. These results highlight the importance of combining flow cytometry with other diagnostic modalities to enhance its diagnostic accuracy.”

Comment #8: 1. Discussion: The discussion could be expanded to include comparisons with existing UTI diagnostic criteria and risk scores.

Response to comment #8: We agreed and expanded the discussion about the differences of our novel UTIRisk score to existing risk score.

Manuscript changes:

• Page 20-21. Discussion. Lines 356-370. “To our knowledge, this is the first risk score that combines urinalysis and flow cytometry parameters to predict UTI in patient with suggested clinical symptoms. Previous UTI risk scores were mainly developed for hospitalized patients, whose characteristics and pathogens differ from our target population.[27-29] While these models showed strong predictive power (C-statistic: 0.79–0.84), they focused on UTIs acquired during hospitalization, often involving antibiotic-resistant pathogens like multidrug-resistant Escherichia coli and Pseudomonas aeruginosa. These scores incorporated demographics (age, sex), laboratory results (blood glucose), medical history, and clinical indicators (Glasgow Coma Score, National Institute of Health Stroke Scale, urinary catheter use) and were designed for prediction and prevention. In contrast, our score prioritizes early diagnosis. In contrast, our risk score is primarily intended for early diagnosis. Notably, only one risk score, developed in England for primary care, aligns with our outpatient setting by focusing on early UTI detection using urinalysis variables.[30]”

Comment #9: - Consider adding a section discussing potential clinical implications of implementing the UTIRisk score in low- and middle-income countries (LMICs) specifically.

Response to comment #9: We agreed and added a section as reviewer suggested.

Manuscript changes:

• Page 21-22. Discussion. Lines 390-403. “Our UTIRisk score has several potential clinical applications, including improving the early diagnosis of UTI, reducing unnecessary urine cultures, and preventing the overuse of antibiotics, which could help mitigate antibiotic resistance, particularly in LMICs. This risk score combines variables from flow cytometry and urinalysis, both of which are quick, easy-to-obtain tests that are inexpensive and require minimal technical expertise. These features enhance diagnostic speed and increase accessibility, making the UTIRisk score particularly valuable in LMICs, where infrastructure and insurance coverage can be challenging. Additionally, the UTIRisk score is user-friendly and can be integrated into mobile applications, allowing physicians to use it quickly and efficiently. Perhaps the most significant advantage of this risk score is that it minimizes reliance on a single diagnostic parameter, such as bacterial count, which can be influenced by microbiome characteristics and socioeconomic factors in different countries. This approach helps address the variability in diagnostic thresholds proposed by previous studies.[13-19]”

Comment #10: 1. Conclusion: The conclusions draw valid implications from the study but should also address the need for further research, possibly emphasizing the types of multicenter studies that would be necessary to validate the UTIRisk score across diverse populations.

Response to comment #10: We agreed and revised the manuscript accordingly.

Manuscript changes:

• Page 23. Conclusion. Lines 431-435. “However, further studies with a multicenter design are needed to validate the efficacy of this diagnostic model across diverse populations, including both LMICs as well as high-income developed countries. Additionally, intervention-based studies, such as randomized controlled trials, are essential before this risk score can be fully integrated into clinical practice.”

Comment #11: 1. References: Ensure that all cited research is current and relevant, particularly concerning comparisons with previous risk scoring models.

Response to comment #11: We double-checked the current references to ensure it contains current and relevant studies.

Comment #12: 2. Data Availability: The authors should clarify how data will be made available upon reasonable request, including what specific data sets may be available.

Response to comment #12: We obtained permission from our hospital to share our data anonymously. The data has been provided as supplemental materials.

Manuscript changes:

• Page 24. Data sharing statement. Lines 448-449. “The data that support the findings of this study are included in the manuscript and supporting information (S Dataset).”

Reviewer #2

Comment #1: Abstract: Background: The background is concise; however, the study aim is not clearly articulated. I wonder if the overall aim of the study can be stated more explicitly.

Response to comment #1: We agreed and tried to state our overall aim more clearly.

Manuscript changes:

• Page 2. Abstract. Lines 28-29. “Our study aims to develop a rapid and reliable predictive model to clinical outcomes.”

Comment #2: Methods: The inclusion period is clearly defined. It may be useful to clarify whether the outpatient department was hospital-based or community-based, as this distinction could affect generalizability—for sure this should be consistent with the main text.

Response to comment #2: We agreed and clarified that the outpatient department was from our hospital.

Manuscript changes:

• Page 2. Abstract. Lines 31-32. “From January 1st to October 31st, 2023, we enrolled patients with symptoms suggesting UTI from the Outpatient Department of our hospital.”

Comment #3: Results: The demographics are well reported. It may enhance clarity to mention whether there were any significant differences in baseline characteristics between the cases and controls.

Response to comment #3: Due to the word limit of 300 in the abstract, we discuss the differences in baseline characteristics between UTI cases and non-UTI controls in the results section and Table 1.

Comment #4: The reported AUC of 0.82 is promising. It might be valuable to briefly add the clinical significance of this value, especially in comparison to existing models or methods.

Response to comment #4: We agree. However, due to the 300-word limit in the abstract, we have addressed the clinical significance of our risk score in relation to existing models in the discussion section.

Manuscript changes:

• Page 20-21. Discussion. Lines 356-370. “To our knowledge, this is the first risk score that combines urinalysis and flow cytometry parameters to predict UTI in patient with suggested clinical symptoms. Previous UTI risk scores were mainly developed for hospitalized patients, whose characteristics and pathogens differ from our target population.[27-29] While these models showed strong predictive power (C-statistic: 0.79–0.84), they focused on UTIs acquired during hospitalization, often involving antibiotic-resistant pathogens like multidrug-resistant Escherichia coli and Pseudomonas aeruginosa. These scores incorporated demographics (age, sex), laboratory results (blood glucose), medical history, and clinical indicators (Glasgow Coma Score, National Institute of Health Stroke Scale, urinary catheter use) and were designed for prediction and prevention. In contrast, our score prioritizes early diagnosis. In contrast, our risk score is primarily intended for early diagnosis. Notably, only one risk score, developed in England for primary care, aligns with our outpatient setting by focusing on early UTI detection using urinalysis variables.[30]”

Comment #5: Introduction: Line 72-73: This point introduces the challenge of antibiotic resistance. Consider briefly mentioning other high-income settings where antibiotic resistance is also a growing concern, making the issue more relevant to a wider audience. Additionally, explaining the impact on healthcare systems in developing countries could provide more depth.

Response to comment #5: We agreed and have added a discussion about the antibiotic resistance in high-income countries and the impact of this condition on healthcare system in developing countries.

Manuscript changes:

• Page 4. Introduction. Lines 78-87. “This rising trend of ABR poses significant challenges, placing additional burden on healthcare systems in affected developing countries, where it has been linked to a 58% rise in crude mortality rates and a 96% increase in ICU admissions.[5] ABR is not restricted to low- and middle-income countries (LMICs); it is also an escalating issue in high-income nations, with projected death rates expected to reach 90.5 per 100,000 by 2050.[6] In Canadian hospitals, the prevalence of ESBL-producing E. coli and K. pneumoniae rose from 3.4% to 7.1% and 1.5% to 4.0%, respectively, between 2007 and 2011.[7] Moreover, in these countries, ABR has been associated with an 84% increase in mortality, an additional 7.4 days of hospital stay, a 49% higher rate of readmissions, and extra costs ranging from US$2,300 to US$29,000.[8]”

Comment #6: Line 73-74: This is a clear and informative point about antibiotic resistance. It would be helpful to clarify the significance of ESBL production for non-specialist readers or consider providing a brief explanation of why this is particularly problematic for UTI management.

Response to comment #6: We agreed and have added the evidence about virulence of ESBL-producing E. coli.

Manuscript changes:

• Page 4. Introduction. Lines 72-74. “Gram-negative bacteria, including Escherichia coli, have become a major concern due to their production of extended-spectrum β-lactamases (ESBLs) which might increase the mortality by more than 3.5 times.[2, 3].”

Comment #7: Line 87: This point adds an important economic perspective. It may be useful to provide a brief comparison of the costs of urine culture versus potential alternatives, helping to contextualize the need for more cost-effective diagnostic methods.

Response to comment #7: We have agreed and included an estimated cost of waste caused by unnecessary urine cultures to emphasize the need for a more cost-effective diagnostic approach.

Manuscript changes:

• Page 5. Introduction. Lines 97-99. “Unnecessary urine cultures with negative results can lead to a waste of

---

## [Decision Letter · Decision Letter 1]

21 Mar 2025

PONE-D-25-03936R1Development of a Novel Risk Score for Diagnosing Urinary Tract Infections: Integrating Sysmex UF-5000i Urine Fluorescence Flow Cytometry with UrinalysisPLOS ONE

Dear Dr. Truyen,

Thank you for submitting your manuscript to PLOS ONE. After careful consideration, we feel that it has merit but does not fully meet PLOS ONE’s publication criteria as it currently stands. Therefore, we invite you to submit a revised version of the manuscript that addresses the points raised during the review process.

We look forward to receiving your revised manuscript.

Kind regards,

Awatif Abid Al-Judaibi, PhD

Academic Editor

PLOS ONE

Journal Requirements:

Reviewers' comments:

Reviewer's Responses to Questions

**Comments to the Author**

1. If the authors have adequately addressed your comments raised in a previous round of review and you feel that this manuscript is now acceptable for publication, you may indicate that here to bypass the “Comments to the Author” section, enter your conflict of interest statement in the “Confidential to Editor” section, and submit your "Accept" recommendation.

Reviewer #2: All comments have been addressed

Reviewer #3: All comments have been addressed

2. Is the manuscript technically sound, and do the data support the conclusions?

Reviewer #2: Yes

Reviewer #3: Yes

3. Has the statistical analysis been performed appropriately and rigorously? 

Reviewer #2: Yes

Reviewer #3: Yes

4. Have the authors made all data underlying the findings in their manuscript fully available?

Reviewer #2: Yes

Reviewer #3: Yes

5. Is the manuscript presented in an intelligible fashion and written in standard English?

Reviewer #2: No

Reviewer #3: Yes

6. Review Comments to the Author

Reviewer #2: Thank you for addressing all of the comments in your revised manuscript.

I have no additional suggestions at this time. However, I would recommend a final proofreading to further enhance the readability of the paper and to correct any remaining grammatical errors.

Reviewer #3: The detailed clarification of the inclusion criteria now effectively addresses concerns about selection bias, clearly delineating the initial diagnosis process. This enhances the manuscript's transparency and robustness. The explanation provided about the selection of retrospective cases, including detailed matching criteria, significantly strengthens the alignment of these cases with the study’s objectives, ensuring that the validation cohort supports the research findings effectively.

7. PLOS authors have the option to publish the peer review history of their article (what does this mean? ). If published, this will include your full peer review and any attached files.

**Do you want your identity to be public for this peer review?** For information about this choice, including consent withdrawal, please see our Privacy Policy .

Reviewer #2: **Yes: ** Mohammad R. Alqudimat

Reviewer #3: No

---

## [Author Response · Author response to Decision Letter 1]

21 Mar 2025

RESPONSE TO REVIEWERS

Reviewer #2

Comment #1: Thank you for addressing all of the comments in your revised manuscript. I have no additional suggestions at this time. However, I would recommend a final proofreading to further enhance the readability of the paper and to correct any remaining grammatical errors.

Response to comment #1: We agreed and thank the reviewer for this comment. We proofread the manuscript and revised to improve the readability.

Reviewer #3

Comment #1: The detailed clarification of the inclusion criteria now effectively addresses concerns about selection bias, clearly delineating the initial diagnosis process. This enhances the manuscript's transparency and robustness. The explanation provided about the selection of retrospective cases, including detailed matching criteria, significantly strengthens the alignment of these cases with the study’s objectives, ensuring that the validation cohort supports the research findings effectively.

Response to comment #1: We thank the reviewer for his comments and suggestion.

Journal Requirements:

Response: We have checked the reference to ensure they are complete and correct.

---

## [Decision Letter · Decision Letter 2]

14 Apr 2025

Development of a Novel Risk Score for Diagnosing Urinary Tract Infections: Integrating Sysmex UF-5000i Urine Fluorescence Flow Cytometry with Urinalysis

PONE-D-25-03936R2

Dear Dr. Thien Tan Tri Tai Truyen,

We’re pleased to inform you that your manuscript has been judged scientifically suitable for publication and will be formally accepted for publication once it meets all outstanding technical requirements.

Kind regards,

Awatif Abid Al-Judaibi, PhD

Academic Editor

PLOS ONE

Reviewers' comments:

Reviewer's Responses to Questions

**Comments to the Author**

1. If the authors have adequately addressed your comments raised in a previous round of review and you feel that this manuscript is now acceptable for publication, you may indicate that here to bypass the “Comments to the Author” section, enter your conflict of interest statement in the “Confidential to Editor” section, and submit your "Accept" recommendation.

Reviewer #2: All comments have been addressed

Reviewer #3: All comments have been addressed

2. Is the manuscript technically sound, and do the data support the conclusions?

Reviewer #2: Yes

Reviewer #3: Yes

3. Has the statistical analysis been performed appropriately and rigorously? 

Reviewer #2: Yes

Reviewer #3: Yes

4. Have the authors made all data underlying the findings in their manuscript fully available?

Reviewer #2: Yes

Reviewer #3: Yes

5. Is the manuscript presented in an intelligible fashion and written in standard English?

Reviewer #2: Yes

Reviewer #3: Yes

6. Review Comments to the Author

Reviewer #2: No further comments/suggestions.

Reviewer #3: I have reviewed the manuscript and commend the authors for their thoughtful revisions. They have adequately addressed the concerns raised in the previous revision trail. I don't have any additional comments or suggestions for this manuscript right now.

7. PLOS authors have the option to publish the peer review history of their article (what does this mean? ). If published, this will include your full peer review and any attached files.

**Do you want your identity to be public for this peer review?** For information about this choice, including consent withdrawal, please see our Privacy Policy .

Reviewer #2: No

Reviewer #3: No

---

## [Editor Report · Acceptance letter]

PONE-D-25-03936R2

PLOS ONE

Dear Dr. Truyen,

I'm pleased to inform you that your manuscript has been deemed suitable for publication in PLOS ONE. Congratulations! Your manuscript is now being handed over to our production team.

Kind regards,

on behalf of

Professor Awatif Abid Al-Judaibi

Academic Editor

PLOS ONE